# Meta-analysis of outcomes from drug-eluting stent implantation in femoropopliteal arteries

**Mingxuan Li**  **\*, Haixia Tu, Yu Yan, Zhen Guo, Haitao Zhu, Jiangliang Niu, Mengchen Yin**

Beijing Fengtai You'anmen Hospital, Beijing, China

\* limingxuan1011@163.com

## Abstract

### Objective

In recent years, studies of drug-eluting stent (DES) for femoropopliteal artery diseases (FPADs) have been gradually published. To explore whether this type of stent is superior to the traditional bare metal stent (BMS), we performed this study.

### Methods

A systematic search for randomized controlled trials (RCTs) in Excerpta Medica Database (Embase), PubMed, Web of Science (WOS), and Cochrane Library was performed on November 29, 2022. We innovatively adopted the hazard ratio (HR), the most appropriate indicator, as a measure of the outcomes that fall under the category of time-to-event data. The HRs was extracted directly or indirectly. Then, the meta-analyses using random effects model were performed. The bias risks of included papers were assessed by the Cochrane Risk of Bias 2.0 tool. This study was registered on the PROSPER platform (CRD42023391944) and not funded.

### Results

Seven RCTs involving 1,889 participants were found. After pooled analyses, we obtained results without propensity on each of the following 3 outcomes of interest: in-stent restenosis (ISR) -free survival, primary patency (PP) survival, and target lesion revascularization (TLR) -free survival (P >0.05, respectively). Because the results of pooled analyses of the other two outcomes of interest (all-cause death free survival and clinical benefit survival) had high heterogeneity both, they were not accepted by us.

### Conclusion

For FPADs, the DES has not yet demonstrated superiority or inferiority to BMS, in the ability to maintain PP, avoid ISR and TLR.

**Data Availability Statement:** All relevant data are within the paper and its Supporting Information files.

**Funding:** The authors received no specific funding for this work.

**Competing interests:** The authors have declared
that no competing interests exist.

# 1. Introduction

Femoropopliteal artery disease (FPAD) is the most common peripheral artery disease (PAD)
[1]. Nowadays, the endovascular-first strategy is now recommended for the majority of symp-
tomatic FPAD patients [2–4]. The complex anatomical and dynamic challenges (such as com-
pression, shortening, torsion) increase the limitations of endovascular treatment strategy [4–
6]. Among numerous endovascular treatment modalities, bare metal stent implantation
(BMSI) is one of the most proven for its superiority, especially in comparison with the percuta-
neous transluminal angioplasty (PTA) i.e., plain old balloon angioplasty (POBA) alone [7–11].
However, the cumulative rate of in-stent restenosis (ISR) at 1 year after BMSI is still ≈30% [7,
9, 12], and that of longer lesions is even ≈50% [12, 13].

After success in the field of coronary artery diseases, drug-eluting stents (DESs) have been
gradually applied to femoropopliteal (FP) artery lesions. In the BATTLE trial, Gouëffic et al.
reported no statistically significant difference in DES implantation (DESI) regarding the ability
to avoid postoperative ISR compared to BMSI (Cox model, P = 0.64) [14]. Whereas in another
randomized controlled trial (RCT) published in the same year, a hazard ratio (HR) with a
value of 3.57 [95% confidence interval (CI), 1.19–10.00] suggesting the superiority on avoiding
ISR of DES was reported by Falkowski et al [15]. A similar condition also occurred between
the other two RCTs. Dake et al. reported in 2016 that DES had better primary patency (PP)
postoperatively compared with BMS (Log-Rank test, P = 0.03) [16]; however, Mori et al.
reported in 2017 that there was no significant difference between them (P = 0.52) [17]. At pres-
ent, there are few meta-analyses on relevant RCTs. In particular, we have not found any meta-
analysis using HR, the most appropriable indicator, as the outcome measure rather than the
odds ratio (OR) or the relative risk (RR), for a variety of the postoperative outcomes of FPADs
that belong to time-to-event data by type [18].

# 2. Materials and methods

## 2.1. Study protocol

According to the PRISMA framework, this study was registered on the PROSPERO platform
(CRD42023391944). The data for analyses were from published RCTs and no new participants
were included. Therefore, no ethical approvals and consent forms of the participants were
required. The PRISMA 2020 checklist is shown in S1 Table.

## 2.2. Search strategy

A systematic search in Excerpta Medica Database (Embase), PubMed, Web of Science (WOS),
and Cochrane Library was performed on December 29, 2022. We searched for all relevant arti-
cles without date limit using "eluting", "stent", "limb" and all possible synonyms of them. The
literature search was independently performed by ML. All entry terms and search commands
can be found in the S1 Fig.

## 2.3. Study selection

We defined the HRs (DESI vs. BMSI) of the following five outcomes of FPADs as the primary
outcome measures of interest: ISR-free survival, PP survival, TLR-free survival, all-cause death
(ACD) -free survival, and clinical benefit (CB) -free survival. Studies that simultaneously met
the following criteria were included by us: 1) the study design was RCT; 2) the language of
publication was English; 3) the operation site was human FP artery suffered from FPAD; 4) the
number of participants in each arm was not less than 10; 5) there are two independent arms,
DESI and BMSI; 6) at least one of the aforementioned HRs was directly reported, or at least

one of the Kaplan-Meier (K-M) survival curves for relevant outcomes was given. The FPAD was defined as a symptomatic disease caused by intraluminal atherosclerotic stenosis or occlusion of the FP artery between inguinal ligament and tibial plateau. We did not limit the definition of the above five outcomes. Studies that did not meet the above criteria or that only included complete duplicates of the outcome data available for extraction were excluded by us.

All the retrieved literature information was imported into the Endnote X9 software, followed by duplicate removal and abstract review. Then, the full texts of all available articles that passed the preliminary screening were downloaded and read to identify those that could be finally included into the study. At the stage of assessing the full texts, the bibliographies and citations were also screened for other potential articles. Two authors (HT and YY) independently performed the study selection. Any discrepancies were resolved by consensus.

## 2.4. Data extraction

After identifying the included studies, we extracted the basic characteristics of the ones and their participants, as well as the primary outcome measures. Directly reported HRs derived from Cox proportional hazard models were preferentially adopted and extracted. If no HRs were found but K-M survival curves were given, we took advantage of the Engauge Digitizer 11.3, an open-source software that can convert graphics to numbers, to transform the information in curves and calculate the HRs [19–21]. Tierney et al. had comprehensively summarized relevant statistical theories and given an HR calculation spreadsheet (Excel format) with preset formula [19]. We used this spreadsheet to calculate HRs, replacing the manual calculation process.

Data extraction was performed by a pair of independent authors (ML and HZ). Any queries and discrepancies were resolved through further discussion to reach a consensus.

## 2.5. Risk of bias assessment

We assessed the bias risks of included RCTs using the Cochrane Risk of Bias 2.0 (RoB 2.0) tool [22]. It evaluates 5 domains: randomization process, deviations from intended interventions, missing outcome data, measurement of the outcome, and selection of the reported results. The risk for each of the 5 domains and overall is described as low, some concerns, or high. The highest risk level among all domains was adopted as the overall assessment result.

Risk of bias assessment was performed by a pair of independent authors (ML and ZG). When the two authors had different opinions on a certain assessment result, the worse one was taken.

## 2.6. Statistical analysis

Stata (Stata Corp., College Station, TX, United States) version 16.0 was used for all statistical analyses. In each meta-analysis, we took the natural logarithms of the extracted HR value and the maximum and minimum values of its 95% CI respectively for each study, and then included the three obtained variables into the "metan" command. To reduce the error, only a random effects model rather than a fixed effects model was taken, regardless of the degree of heterogeneity among studies [23]. And we corrected the degree of freedom by the restricted maximum likelihood estimation [24]. For any hypothesis test, only the results with a P value less than 0.05 could be considered statistically significant. All pooled analyses were performed independently by ML.

The formula of the Cox proportional hazard model (1) and those for meta-analysis based on extracted or transformed data (2, 3) are shown as follows [19, 20, 25]:

$$h(t, X) = h_0(t) exp(\beta_1 x_1 + \beta_2 x_2 + \cdots + \beta_m x_m) \tag{1}$$

$$pooled\ lnHR = \sum \left(\frac{lnHR}{V}\right) / \sum \left(\frac{1}{V}\right) \tag{2}$$

$$V = [\ln(upper\ 95\%CI) - \ln(lower\ 95\%CI)]^2 / (2*1.96)^2 \tag{3}$$

## 2.7. Heterogeneity assessment and sensitivity analysis

The heterogeneity across the studies was assessed and reported as a percentage using the $I^2$ index value and as a P value using the Cochrane Q test of $\chi^2$ [26, 27]. $I^2 < 50\%$ suggests a low heterogeneity, and $I^2 \geq 50\%$ suggests high. P $<0.1$ for Q test suggests a low heterogeneity, and P $\geq 0.1$ suggests high. Only the models with low heterogeneity suggested by both tests were adopted.

After obtaining a model with low heterogeneity that was included at least 3 studies, the checking calculation was performed by omitting the included studies one by one, to analyze the sensitivity of the model. When a new pooled effect size (ES), i.e., ln (HR) value, which was far away from the ES obtained previously or even beyond the 95% CI range, was obtained from the meta-analysis after omitting one study, this study was considered to bring about instability. In initial plan, once a meta-analysis model had a high heterogeneity and $\geq 4$ studies had been included, we would try to find out the variables that cause heterogeneity by meta regression or empirically, and conduct subgroup meta-analyses according to them.

The above works were performed by a pair of independent authors (HZ and ZG). When the two authors had different opinions on a certain assessment result, the worse one was taken.

## 2.8. Publication bias assessment

The following methods of publication bias assessment were performed only if no fewer than 3 studies were included in each meta-analysis. One side, the Egger's test was used [28]. P $<0.05$ means a high publication bias. And a funnel plot also be drawn [29]. An apparently asymmetric plot with the ES value as axis reflects a high publication bias. The assessment was performed by a pair of independent authors (JN and MY). When the two authors had different opinions on a certain assessment result, the worse one was taken.

## 2.9. Evidence quality grade assessment

After finishing all meta-analyses, we used the GRADE (Grading of Recommendations Assessment, Development and Evaluation) system to evaluate the qualities of evidence and make recommendations [30]. The evidence derived from meta-analyses is all initially set to high, and respectively graded as high, moderate, low, or very low, finally. Its rating will be lowered by a corresponding number of levels once it appears suspect in terms of overall bias, publication bias, inconsistency, imprecision, or indirectness.

The assessment was performed by a pair of independent authors (ML and HT). When an assessment result was discordant and consensus could not be reached, the one with lower grade was adopted.

## 3. Results

### 3.1. Studies and data

We initially identified 1,234 articles by searching 4 academic databases, of which 507 were evaluated after removing duplicates. And 23 articles were retained after the title abstract sieve. After reviewing the full text, 7 RCTs were finally included in this study [14–17, 31–33]. Among them, one [31] reported the pooled, longer-term follow-up results from 2 periods [34, 35] of the SIROCCO trial. The PRISMA flowchart of study selection is shown in Fig 1.

The publication years of the 7 included RCTs ranged from 2006 to 2022 [31, 33]. There was only one paper was from a single center rather than multiple centers [15]. Three papers only included de novo FP lesions [14, 17, 32], two also included restenosis lesions [16, 31], and two did not mention this aspect of pathological characteristics [15, 33]. The arms in each study were independent with no patient conversions, and the stents implantations were all primary, that is, regardless of whether PTAs had been performed previously or their effects, except for 1 paper [16]. Unlike others, the control arm in the original trial design of this study (Zilver PTX) was PTA with or without BMSI. Among the patients in this arm, those who underwent acute PTA failure were further divided into two subgroups for comparison: DESI and BMSI [16]. This kind of stent implantation was considered as secondary and provisional. The investigators reported data available on outcomes of this subset. To avoid the inclusion of patients who underwent PTA alone, we adopted only the above patients in this study (S2 Fig).

A total of 939 patients underwent DESI and 950 underwent BMSI were included, respectively. Their mean age exceeded at least 65y (from reported data), and their majority was male. More specific main characteristics of these studies and baseline data of patients are shown in

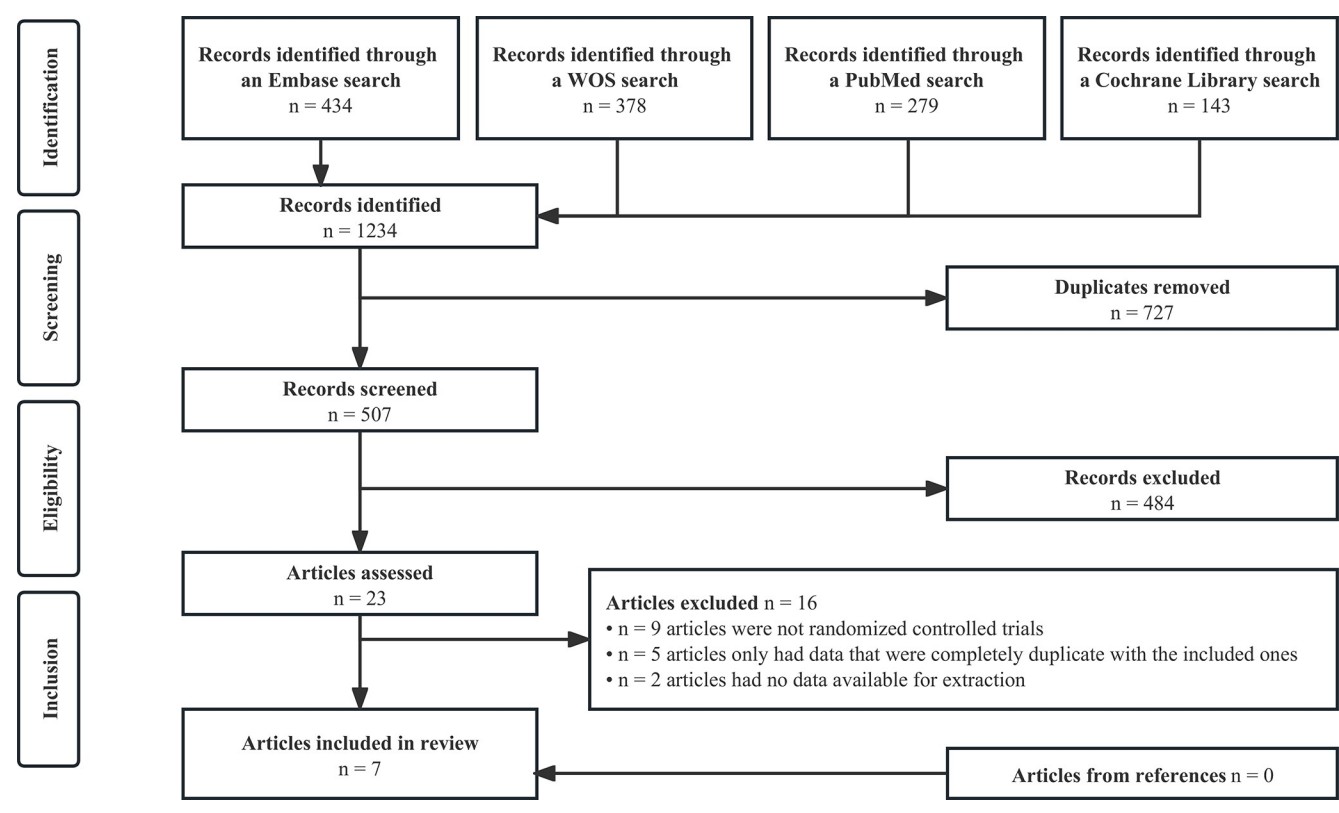

**Fig 1. PRISMA flowchart.**

Table 1. Except for those not explicitly mentioned, the definitions of outcomes of interest in the included studies were close. The specific definitions are shown in Table 2.

### 3.2. Risks of bias

After assessment, the overall risk of bias in 2 included studies was "high" [16, 32], in four it was "some concerns" [14, 15, 17, 31], and in one it was "low" [33]. The detailed final assessment results are shown in Table 3.

### 3.3. ISR-free survival

Two studies [14, 15] reported the HRs directly and another [32] gave the K-M survival curve. The result of one [15] was particularly in favor of DESI (HR = 3.57, P = 0.023), whereas those of 2 others [14, 32] were not (P >0.05). The obtained model had a low heterogeneity ($I^2$ = 34.6%, P = 0.217 for Q test), and suggested that there was no statistical difference between the two arms in the risk of postoperative ISR [HR = 1.51 (95% CI, 0.99–2.31), P = 0.058] (Fig 2A). The stability of the model was satisfactory (Fig 2B). The P derived by Egger's test which value was 0.222 and the visual symmetrical funnel plot, both suggested a low publication bias of the model (Fig 2C).

### 3.4. PP survival

Three studies [16, 17, 33] gave the K-M survival curves and another [14] reported the HR directly. The result of one [16] of them was particularly in favor of DESI (P = 0.003), whereas those of the other three were not (P >0.05). After pooling those of these 4 studies, we obtained a result without propensity (P = 0.941). However, the high $I^2$ value (66.3%) and the low P value (0.031) for Q test jointly suggested that the model was highly heterogeneous, so it was not accepted by us (Fig 3A). After omitting 3 studies separately, we obtained only one model which had lowest heterogeneity ($I^2$ = 45.0%, P = 0.163 for Q test). The result remained nonsignificant in favor of either DESI or BMSI, that is, HR = 1.28 (95% CI, 0.79–2.06) and P = 0.312 (Fig 3B). We could not perform sensitivity analysis and assessment of publication bias due to too few included studies. The stability of the model was satisfactory (Fig 3C). The Egger's test yielded a P value of 0.334, and the funnel plot also appeared symmetrical, both suggesting a low publication bias of the obtained model (Fig 3D).

### 3.5. TLR-free survival

One study that directly reported the HR [14] and other two that gave the K-M survival curves [16, 31] were included in a meta-analysis. All the results of these 3 studies did not support a statistical difference in postoperative TLR risk between BMSI and DESI (P >0.05). So without surprise, the pooled analysis derived a same result [HR = 1.66 (95% CI, 0.93–2.96), P = 0.089] (Fig 4A). An $I^2$ value of 0.0% and a P value for Q test of 0.608 suggested low heterogeneity of the derived model (Fig 4A). Sensitivity analysis suggested high stability of this model (Fig 4B). The model had P = 0.799 for Egger's test and its funnel plot was roughly symmetrical, which suggested no high publication bias (Fig 4C).

### 3.6. Survival of ACD-free and CB

There were 2 studies with available data in terms of ACD-free survival. One [14] of them directly reported an HR without propensity (7.3, 95% CI 0.9–59.3), and the other [15] gave a relevant K-M curve with same suggestion (P = 0.576). We pooled the results of the two studies but obtained a highly heterogeneous model ($I^2$ = 75.0%, P = 0.045 for Q test), so we did not

Table 1. Main characteristics and baselines of the included studies.

| Author | Registry | Enrollment period | Recruited RCs | Recruited lesions | DES | Control BMS | Runoff count | Primary stenting‡ | Num of patients | Mean age (y) | Male (%) | CLI (%) | CTO (%) | Restenosis (%) | Mean LL (mm) | Follow up period (y) |
|---|---|---|---|---|---|---|---|---|---|---|---|---|---|---|---|---|
| Duda et al. (2006) | SIROCCO | 2001–2002 | 1–4 | FA | Sm | Sm | NI | Y vs. Y | 47 vs. 46 | 66 vs. 66 | 66 vs. 78 | NI | 69 vs. 57 | 11 vs. 4 | 85 vs. 81 | 2 |
| Dake et al. (2016) | Zilver PTX | 2005–2008 | 2–6 | FPA | ZP | Z | >1 | Y vs. N | 61 vs. 59 | 68 vs. 68 | 66 vs. 64 | 9 vs. 8 | 33 vs. 27 | 5 vs. 6 | 66 vs. 63 | 5 |
| Mori et al. (2017) | Kanagawa PTA | 2014 | NI | FA | ZP | Z/Sm/M | NI | Y vs. Y | 27 vs. 279 | NI | 67 vs. 72 | 15 vs. 33 | NI | 0 vs. 0 | NI | 1 |
| Miura et al. (2018) | DEBATE | 2014–2016 | 2–4 | FPA | ZP | M | NI | Y vs. Y | 84 vs. 84 | 73 vs. 73 | 71 vs. 65 | 7 vs. 12 | 51 vs. 36 | 0 vs. 0 | 110 vs. 96 | 1 |
| Gouëffic et al. (2020) | BATTLE | 2014–2016 | 2–5 | FPA | ZP | M | >1 | NI | 86 vs. 85 | 71 vs. 68 | 72 vs. 73 | 21 vs. 18 | 38 vs. 35 | 0 vs. 0 | 69 vs. 76 | 2 |
| Falkowski et al. (2020) | NI | 2016 | 2–5 | FPA | ZP | Z | >1 | Y vs. Y | 126 vs. 130 | 67 vs. 65 | 64 vs. 63 | NI | 40 vs. 48 | NI | 94 vs. 128 | 3 |
| Gouëffic et al. (2022) | EMINENT | 2016–2020 | 2–4 | FPA | El | I/Su/L/Ev† | >1 | Y vs. Y | 508 vs. 267 | 69 vs. 69 | 72 vs. 67 | 4 vs. 3 | 42 vs. 40 | NI | 76 vs. 72 | 1 |

Representation of variables with data for double arms: DES arm vs. control arm. NI, no information; RC, Rutherford classification; FA, femoral artery; FPA, femoropopliteal artery; DES, drug-eluting stent; Sm, SMART; ZP, Zilver PTX; El, Eluvia; BMS, bare metal stent; Z, Zilver; M, Misago; I, Innova; Su, Supera; L, Lifestent; Ev, Everflex; Num, number; CLI, critical limb ischemia; CTO, chronic total occlusion; LL, lesion length. †Stents accounting for less than 10% are not listed. ‡DES implantation was performed regardless of whether predilatation was performed or the outcome of predilatation

**Table 2. Definitions of the outcomes during follow up extracted from the included studies.**

| Author | ISR | PP | TLR | ACD | CB |
|---|---|---|---|---|---|
| Duda et al. (2006) | NA | NA | NI* | NA | NA |
| Dake et al. (2016) | NA | < 50% stenosis via duplex ultrasonography (with PSVR < 2.0) or from arteriography* | Reintervention for ≥ 50% stenosis within ± 5 mm of the target lesion after recurrent clinical symptoms* | NA | Freedom from persistent or worsening symptoms of ischemia* |
| Mori et al. (2017) | NA | The target lesion did not undergo TLRs and remained patency* | NA | NA | NA |
| Miura et al. (2018) | The PSVR > 2.0 detected by doppler ultrasound at target lesion* | NA | NA | NA | NA |
| Gouëffic et al. (2020) | The PSVR > 2.4 detected by doppler ultrasound at target lesion | NI | NI | All-cause death | A sustained upward shift of 1 category of the RC for patients with claudication and by wound healing and rest pain resolution for patients with CLI, without the need for repeated TLR |
| Falkowski et al. (2020) | > 50% stenosis seen via duplex ultrasonography (with PSVR > 2.0) or from CT scan | NA | NA | All-cause death* | NA |
| Gouëffic et al. (2020) | NA | PSVR ≤ 2.0 detected by doppler ultrasound without any TLRs | NA | NA | NA |

ISR, in stent restenosis; NA, not available (no data extracted); PSVR, peak systolic velocity ratio; CT, computer tomography; PP, primary patency; NI, no information (data available); TLR, target lesion revascularization; ACD, all-cause death; CB, clinical benefit; RC, Rutherford classification; CLI, critical limb ischemia. *Data extracted from survival curve presented in the article and thus transformed.

accept it (Fig 5A). There were similarly only 2 studies with available data on CB survival: one [14] directly reported an HR without propensity (0.9, 95% CI 0.5–1.6) and the other [16] gave a K-M curve in favor of DESI (P = 0.02). The result of pooled analysis was similarly highly heterogeneous ($I^2$ = 65.7%, P = 0.088 for Q test) and not accepted by us (Fig 5B).

**Table 3. Risk bias assessment results of included studies.**

| The domains in RoB 2.0[†] for RCTs | Duda et al. (2006) | Dake et al. (2016) | Mori et al. (2017) | Miura et al. (2018) | Gouëffic et al. (2020) | Falkowski et al. (2020) | Gouëffic et al. (2022) |
|---|---|---|---|---|---|---|---|
| 1. Randomisation process | Low | Low | Low | High | Low | Some concerns | Low |
| 2. Deviations from the intended interventions | Low | High | Low | Low | Low | Low | Low |
| 3. Missing outcome data | Some concerns | Some concerns | Some concerns | Low | Some concerns | Low | Low |
| 4. Measurement of the outcome | Low | Low | Low | Low | Low | Low | Low |
| 5. Selection of the reported result | Low | Low | Low | Low | Low | Low | Low |
| 6. Overall | Some concerns | High | Some concerns | High | Some concerns | Some concerns | Low |

RoB, risk of bias tool; RCT, randomized controlled trial. [†]The risk is graded into 3 levels: low, some concerns, and high.

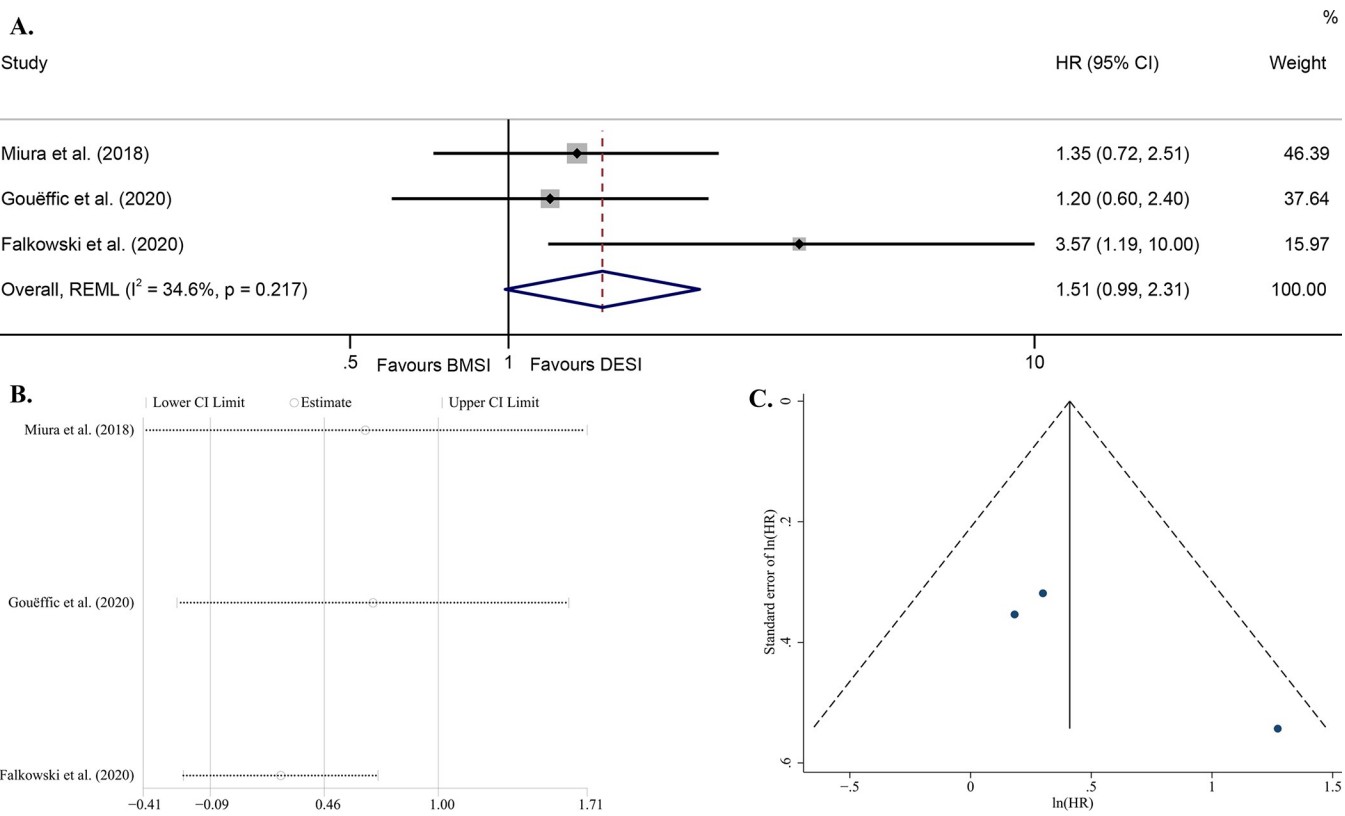

**Fig 2.** Forest plot of ISR-free survival (random effects model, P = 0.058) (A); Sensitivity analysis of the model assuming that each study is omitted separately [ln (HR)] (B); Funnel plot with pseudo 95% confidence limits (C). Abbreviations: HR, hazard ratio; CI, confidence interval; REML, restricted maximum likelihood; BMSI, bare metal stent implantation; DESI, drug-eluting stent implantation; ISR, in-stent restenosis.

### 3.7. Evidence quality grade

By pooling the results of RCTs, we obtained 3 corollaries that could be recommended. They revealed that the DESI did not demonstrate a statistically advantage or disadvantage compared to BMSI in terms of ISR-free survival, PP survival, and TLR-free survival for FPADs. After assessment according to the GRADE criteria, their recommendation ratings were all lowered to "low" due to the shortcomings both of "study limits" and "evidence indirectness". Details are detailed in Table 4.

## 4. Discussion

This study systematically reviewed and analyzed multiple follow-up outcomes of DESI performed in FP arteries by including 7 RCTs. The results show that, compared to the traditional BMSI, there were no clear superiority or inferiority of DESI in terms of ISR-free survival, PP survival, and TLR-free survival.

For FP lesions, the recognized disadvantage of PTA alone is a high rate of restenosis and concomitant need for TLR [4]. Unlike the coronary, PAD involves longer segments, often at multiple levels with decreased flow rates leading to restenosis even when immediate angiographic results are excellent [11]. To address this limitation, BMSs began to be applied in FPADs in the 1980s [4]. The BMS, represented by nitinol stent, can provide radial support and compression resistance, resulting in extended patency of target lesions [7]. However, stents left in the vascular lumen have brought new problems. The core one is ISR, which reduces the PP

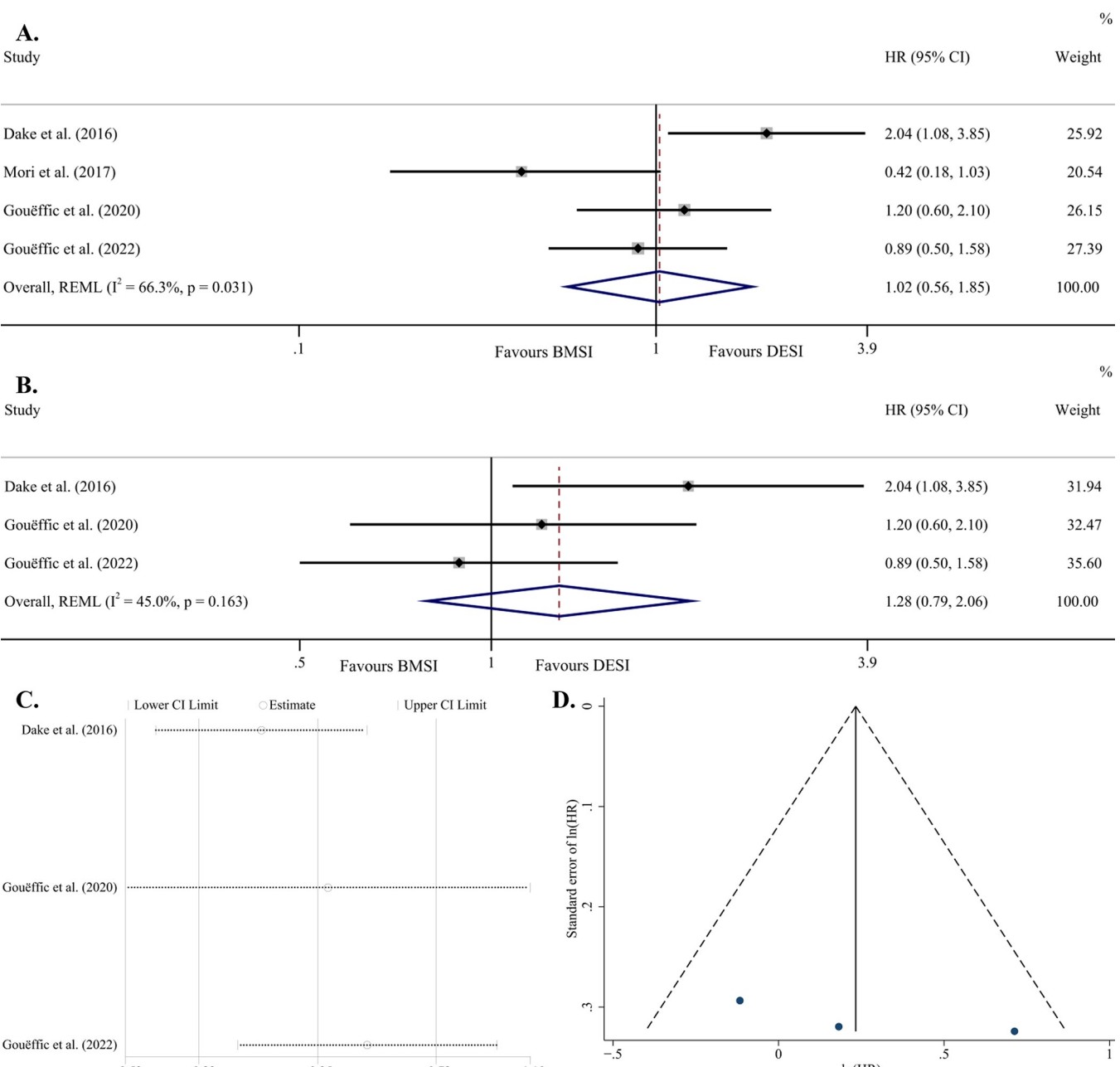

**Fig 3.** Forest plot of PP survival (random effects model with high heterogeneity) (A); Forest plot of PP survival (random effects model with low heterogeneity, P = 0.312) (B); Sensitivity analysis of the model assuming that each study is omitted separately [ln(HR)] (C); Funnel plot with pseudo 95% confidence limits (D). Abbreviations: HR, hazard ratio; CI, confidence interval; REML, restricted maximum likelihood; BMSI, bare metal stent implantation; DESI, drug-eluting stent implantation; PP, primary patency.

and relapses clinical manifestations of ischemia, thus requiring the TLR. Even many studies have reported that the cumulative ISR rate at the first postoperative year after BMSI for FPADs had exceeded 30% [7, 9, 36, 37].

Proliferation of vascular smooth muscle cells (SMCs) is a significant cause of neointimal hyperplasia, which ultimately causes the ISR [38, 39]. The anti-proliferation mechanism of some drugs such as paclitaxel, sirolimus, etc. is theoretically beneficial to antagonize the

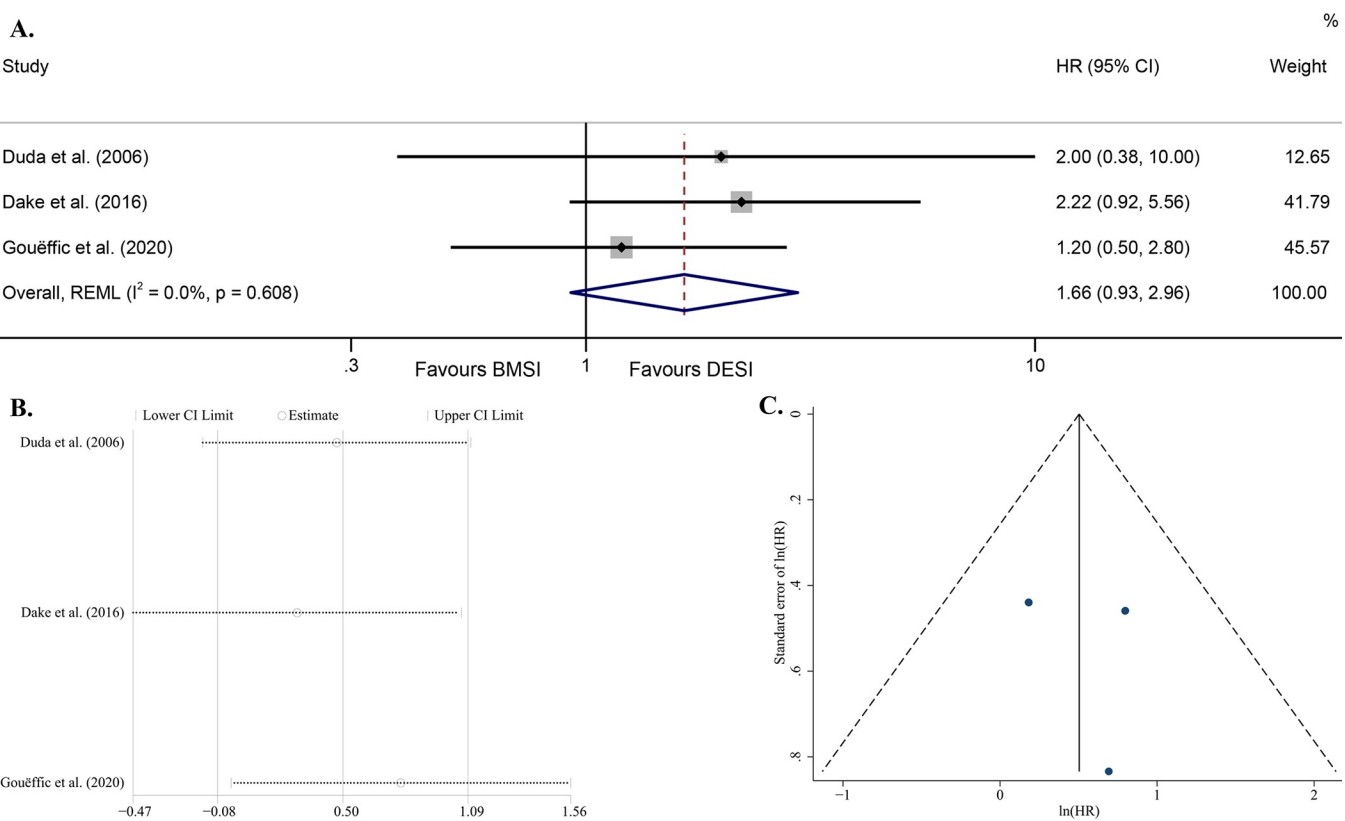

**Fig 4.** Forest plot of TLR-free survival (random effects model, P = 0.089) (A); Sensitivity analysis of the model assuming that each study is omitted separately [ln (HR)] (B); Funnel plot with pseudo 95% confidence limits (C). Abbreviations: HR, hazard ratio; CI, confidence interval; REML, restricted maximum likelihood; BMSI, bare metal stent implantation; DESI, drug-eluting stent implantation; TLR, target lesion revascularization.

proliferation of SMCs and reduce the ISR rate. The superiority of stents eluted with the above drugs for coronary arteries has been extensively demonstrated in previous studies [40–43], even for infrapopliteal arteries with similar diameters [44]. It is no surprise that such promising stents have gradually been used in FP lesions. In a retrospective study published in 2016 [45], Soga et al. reported that there were no statistically significant differences (P >0.05) all between DES and BMS on the cumulative incidence of 1-year TLR, major amputation, and other outcomes for FPADs. However, Meng et al. reported in a retrospective study in 2018 that DES had a lower 2-year cumulative ISR rate (0% vs. 23%, P = 0.009) [46]. In recent years, although there have been some meta-analyses on outcomes of our interest, they all used OR as the outcome measure rather than the more appropriate HR [47, 48]. In order to include latest studies and obtain more convincing results through more scientific analysis methods, we have performed this systematic review and meta-analysis of the published RCTs.

There is no doubt that the outcomes of interest in the presented study are time-to-event data all, that is, time is a significant variable that affects the occurrence of outcomes [18]. Because there is no way for the pathophysiological process of atherosclerosis to be completely terminated, restenosis will necessarily occur. So over time, the rate of outcomes such as ISR will gradually increase. For example, for a cohort of FPAD patients, the cumulative ISR rate at the first postoperative month will not be the same as that at the 10th postoperative year. A more straightforward explanation is that a patient will surely die but not necessarily within 1 year postoperatively. Among the numerous indicators including more common OR and RR,

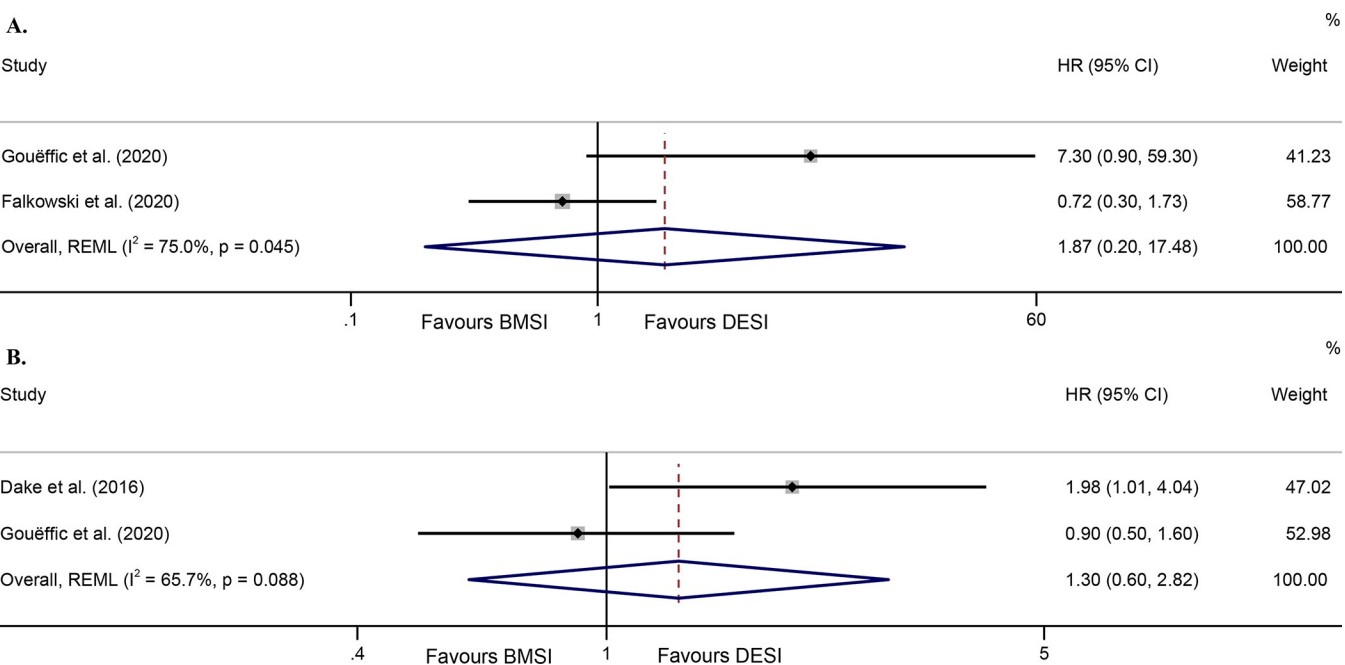

**Fig 5.** Forest plot of ACD survival (random effects model with high heterogeneity) (A); Forest plot of CB survival (random effects model with high heterogeneity) (B). Abbreviations: HR, hazard ratio; CI, confidence interval; REML, restricted maximum likelihood; BMSI, bare metal stent implantation; DESI, drug-eluting stent implantation; ACD, all-cause death; CB, clinical benefit.

HR is the most appropriate measure for the time-to-event outcomes [18, 19]. Another advantage of this is that it is not necessary to consider the time variable, so when selecting which studies would be included in a pooled analysis on a certain outcome, it is no longer limited by the follow-up period, which is helpful to increase the number of included studies. Therefore, we decided to adopt the HR as outcome measure in this study. But a followed practical problem was that RCTs reporting the HRs directly might be few. Fortunately, most of the retrieved RCTs gave the survival curves after analyses of some outcomes, which provided the possibility to extract data. We used the software to extract the survival probabilities at different follow-up moments, and then the HR value with its 95% CI of a certain outcome was derived according

**Table 4. Evidence quality grade assessment of pooled outcomes of interest.**

| Outcome | Source of data | Num of participants | Pooled HR (DES vs. BMS) | Certainty of the evidence (GRADE)[†] | Reasons for lowering the rating |
|---|---|---|---|---|---|
| ISR-free survival | 3 RCTs | 595 | 1.51 (95% CI, 1.00–2.30) | Low | SL, EI |
| PP survival | 3 RCTs | 1066 | 1.28 (95% CI, 0.56–1.85) | Low | SL, EI |
| TLR-free survival | 3 RCTs | 384 | 1.66 (95% CI, 0.93–2.96) | Low | SL, EI |

ISR, in stent restenosis; PP, primary patency; TLR, target lesion revascularization; RCT, randomized controlled trial; Num, number; HR, hazard ratio; DES, drug-eluting stent; BMS, bare metal stent; CI, confidence interval; GRADE, grading of recommendations assessment, development and evaluation; SL, study limitations; EI, evidence indirectness. [†]High certainty: we are very confident that the true effect lies close to that of the estimate of the effect; moderate certainty: we are moderately confident in the effect estimate; low certainty: our confidence in the effect estimate is limited; very low certainty: we have very little confidence in the effect estimate.

to formulas. This method of deriving HR value has high accuracy and stability, and has been widely recognized [19–21]. We think that this may be the first meta-analysis to do so in relevant area.

Pooled analysis suggested no significant difference between the two treatment modalities in their ability to maintain postoperative ISR-free survival for FPADs (P = 0.058). Unlike the results without propensity of the other 2 studies, that of 1 study favored DESI strongly [HR = 3.57 (95% CI, 1.19–10.00)] [15], which led to a result with significant difference almost in the final analysis. The range of 95% CI of HR obtained in this study was wide (1.19–10.00), which led to the reduction of the accuracy of this result, thus making its weight in the pooled analysis small (16%). We note that the mean length of treated target lesions in the BMSI arm in this study was 128 mm, which was subjectively significantly larger than those in the other 2 studies (96 mm and 76 mm). There is indeed evidence that the restenosis rate after BMSI is significantly higher, or even more than 50% at 1st postoperative year, for long FP lesions rather than short ones [12, 13]. However, in a prospective non-randomized trial published in 2017 specifically for long (>15cm) FP lesions, there was also no significant difference between the two treatment modalities (P = 0.62 for Log-Rank test) [49]. Since most of the definitions of PP in the included studies are not the opposite of ISR simply, and even some [14] had not given exact explanation, we regarded the PP survival as an independent outcome. After the three inconsistent but low heterogeneity results were pooled, the derived result suggested that there was no difference between the two stents in the ability to maintain postoperative PP survival. In this analysis, unlike the other two trials, the Zilver PTX trial [16] reported a result favoring DES (P = 0.003). The initial grouping of this study at the design stage was primary DESI versus PTA (with or without different types of stent implantation); however, various subgroups were also analyzed in the analysis stage. The comparison result we could extract were only from the "secondary DESI versus secondary BMSI, after acute PTA failure". This suboptimal stent implantation was the most noticeable difference between this study and the others. Theoretically, DES seems to be more useful for the immediate arterial dissection after PTA than BMS. But unfortunately, we have not found any other RCTs or even other controlled studies that reported similar comparisons. The result of the pooled analysis of TLR survival were not surprising, because the results of the 3 studies included were all without propensity (P >0.05). Notably, the Zilver PTX trail [16] reported a PP survival outcome in favor of DES; but this scenario was not seen with respect to TLR-free survival. This seems to indicate that the TLR is not inevitable after the occurrence of ISR. Many studies have confirmed the benefits of persistent physical exercise on improving quality of life, especially on extending walking distance, so as to avoid TLR [50–52]. We believe that this may be related to its ability to enhance blood circulation compensation rather than just maintain the patency of the target lesion. Of course, this may need to be supported by more evidence. Regrettably, in terms of ACD-free survival and CB survival, due to the high heterogeneity of the studies included in the analyses, the results obtained were not accepted by us. We look forward to more studies.

The longer lesions have been identified as an independent predictor of restenosis [53]. In the previously mentioned prospective study specifically for the long FP lesions [49], not only the ISR-free survival, but also all the other reported outcomes have suggested no statistical differences between the two treatment modalities. Other relevant controlled studies were not retrieved. Several other studies reported a cumulative ISR rate of 30–40% and a cumulative TLR rate of 19–21% at 1st postoperative year of DESI for above lesions [54, 55]. This is clearly no better than the data in one of our included studies, in which the mean length of the included lesions in the BMSI arm was 128 mm and the above cumulative rates at 3rd postoperative year were 35.4% and 31.5%, respectively [15]. In conclusion, the available evidence does not support a superiority of DES compared with BMS for long FP lesions. Another noteworthy

special type of FP lesion is the ISR. BMSI is not the accepted default treatment modality for this disease [56, 57]. And we have not found any study specifically comparing DES and BMS in this area. In Zilver PTX single-arm trail, the treatment of 119 FP-ISR lesions with DES had an estimated PP rate of 78.8% at 1st postoperative year [58]. And Tomoi et al. reported that the estimated freedom from recurrent ISR (ReISR) at 2nd DESI postoperative year was 79% in a study published in 2016 [59]. These results are even very close to those of BMSI for de novo FP lesions reported in our included studies [14, 32]. Therefore, we believe that the postoperative patency of DES is better than that of BMS, for FP-ISR lesions.

In recent years, the fluoropolymer-based paclitaxel-eluting stent, Eluvia™, described as "the new generation of DES", has been more and more widely used in clinic. Theoretically, the polymer on this type of stent guarantees longer drug elution period comparing to the polymer-free DES (represented by Zilver® PTX®, that accounted for the majority of the DESs in the included studies), and thus enables the drug to function longer [60]. Moreover, the good biocompatibility of this polymer may make the stent safer than the other early polymer-coated DESs [31, 61–63]. A prospective one-arm trail, CAPSICUM [64], reported a 13% cumulative ISR rate at 1st postoperative year of Eluvia implantation for FP lesions, which seems to be significantly better than the 28% reported by DEBATE trail [32]. A subgroup study from the IMPERIAL RCT [65] reported a 1-year cumulative PP rate of 87%, which was similar to that reported by the Kanagawa PTA trail (84%) [17]. But it is important to note that the former was specifically for long lesions (>140 mm), and the result appears to be better than that of the aforementioned study on the Zilver PTX DES (52%) [39]. Another prospective one-arm trail, MAJESTIC [66], reported a 1-year estimated PP rate of 84%, which also seems to be significantly better than the 75% reported by the BATTLE trail [14]. The results of studies in which two types of DES, Eluvia and Zilver PTX, were directly compared were also similar. Eluvia performed better in both PP survival and TLR-free survival in the IMPERIAL trial (1-year follow-up period, P <0.02) [60]. Soga et al. reported in 2019 that the cumulative mean late lumen loss and ISR rate in the Eluvia arm were lower at 1st postoperative year (P <0.02), but the superiority on cumulative TLR rate was not yet significant (0% vs. 8%, P = 0.08) [67]. And Gray et al. reported in 2022 that, Eluvia was more effective and less costly than Zilver PTX [68]. In 2022, the 1-year results of the EMINENT trial directly comparing DES and BMS were published [33]. Except for that of PP (83% vs 74%, P <0.01), there was no intergroup difference in the 1-year cumulative rates of all outcomes (P >0.05); Moreover, after data conversion and extraction, we found no difference in PP survival between groups (P = 0.312). This has beat to our confidence in Eluvia DES. We look forward to more RCTs about this novel DES.

This presented study has some limitations. First, the number of studies included in each analysis was all limited, which reduced the persuasiveness of the study. If the numbers of included studies (i.e., the sample size) increased, some results in favor of DES might be obtained. Second, although the image information extraction technology was used to increase the sample size and the data conversion was automatically finished with software, the obtained data had been reprocessed after all, which may reduce the statistical validity to a certain extent. Third, due to some objective reasons, the corollaries derived from analyses were all "low", which was the penultimate level after GRADE grading, reducing the credibility of recommendations.

## 5. Conclusions

The results of this study show that, for FPADs, the DES has not yet demonstrated superiority or inferiority to BMS, in the ability to maintain PP, avoid ISR and TLR.

## Supporting information

**S1 Table. PRISMA checklist.**
(PDF)

**S1 Fig. Search terms used in the literature search on different platforms.**
(TIF)

**S2 Fig. Patient selection during data extraction from the Zilver PTX trial [16].**
(TIF)

## Author Contributions

**Conceptualization:** Mingxuan Li.

**Data curation:** Mingxuan Li, Haitao Zhu.

**Methodology:** Mingxuan Li, Zhen Guo, Jiangliang Niu, Mengchen Yin.

**Resources:** Mingxuan Li, Haixia Tu, Yu Yan.

**Software:** Mingxuan Li.

**Supervision:** Mingxuan Li.

**Validation:** Haixia Tu, Yu Yan.

**Writing – original draft:** Mingxuan Li.

**Writing – review & editing:** Haixia Tu.

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
