## [Decision Letter · Decision Letter 0]

1 Aug 2023

PONE-D-23-21377

Meta-analysis of outcomes from drug-eluting stent implantation in femoropopliteal arteries

PLOS ONE

Dear Dr. Li,

Thank you for submitting your manuscript to PLOS ONE. After careful consideration, we feel that it has merit but does not fully meet PLOS ONE’s publication criteria as it currently stands. Therefore, we invite you to submit a revised version of the manuscript that addresses the points raised during the review process.

We look forward to receiving your revised manuscript.

Kind regards,

Redoy Ranjan, MBBS, MRCSEd, Ch.M., MS (CV&TS), FACS

Academic Editor

PLOS ONE

Journal Requirements:

Reviewers' comments:

Reviewer's Responses to Questions

**Comments to the Author**

1. Is the manuscript technically sound, and do the data support the conclusions?

Reviewer #1: Yes

Reviewer #2: Yes

Reviewer #3: Partly

2. Has the statistical analysis been performed appropriately and rigorously? 

Reviewer #1: Yes

Reviewer #2: Yes

Reviewer #3: I Don't Know

3. Have the authors made all data underlying the findings in their manuscript fully available?

Reviewer #1: Yes

Reviewer #2: Yes

Reviewer #3: Yes

4. Is the manuscript presented in an intelligible fashion and written in standard English?

Reviewer #1: No

Reviewer #2: Yes

Reviewer #3: Yes

5. Review Comments to the Author

Reviewer #1: This meta-analysis is an interesting topic in vascular surgery, since the debate on the DEC, BMS, DCB, et al.

This reviewer has several major concerns:

1, As the author stated, the length of the disease is critical for the outcome, length should be analyzed.

2, Debate on the popliteal artery stent implantation is still ongoing because of the anatomy, is the length of disease of the popliteal artery a risk factor?

3, The short and long term improvement of both therapies.

4, What are the risk factors that impact the outcome?

5, Runoffs are critical to the endovasular therapy, the condition of the runoffs.

6, tyro errors should be corrected. like "2000".

minor:

1, Page 3, Line 58-63; Are there any other criteria for inclusion in the population? For example, age, gender, etc.

2. Page 5, Line 143-145; Please add additional documentation to support this.

3. Page 9, Line 209; Please add literature to support that PTA alone has the disadvantage of a high restenosis rate, as well as the need for a TLR

4. Page 3，Line65，What specifically is meant by the failure to qualify the ending? What are the criteria for determining re-narrowing? What are the inclusion criteria for recanalization? Do different inclusion criteria affect the results?

5. Some of the results used data from only two articles, and that some of the data were extracted from the KM curves? Do the authors have a solution for this?

6. Page 11，line 306 Did the authors consider including non-RCT results from "the new generation of DES" to see if outcomes changed significantly?

Reviewer #2: Well done, nice workup and nice well organized results. You did a rigorous point by point review .In my opinion it deserves very wide exposure for a maximum benefit for specialties working in this filed.

Reviewer #3: Thank the authors for the opportunity to read their work.

Please see below are a series of comments for your consideration.

It would be appropriate to include the bibliographic search terms within the supplementary material.

There are randomized clinical trials that have not been included in the analysis, such as:

Gouëffic Y, Torsello G, Zeller T, Esposito G, Vermassen F, Hausegger KA, Tepe G, Thieme M, Gschwandtner M, Kahlberg A, Schindewolf M, Sapoval M, Diaz-Cartelle J, Stavroulakis K; EMINENT Investigators. Efficacy of a Drug-Eluting Stent Versus Bare Metal Stents for Symptomatic Femoropopliteal Peripheral Artery Disease: Primary Results of the EMINENT Randomized Trial. Circulation. 2022 Nov 22;146(21):1564-1576. doi: 10.1161/CIRCULATIONAHA.122.059606. Epub 2022 Oct 18. PMID: 36254728.

Spreen MI, Martens JM, Hansen BE, Knippenberg B, Verhey E, van Dijk LC, de Vries JP, Vos JA, de Borst GJ, Vonken EJ, Wever JJ, Statius van Eps RG, Mali WP, van Overhagen H. Percutaneous Transluminal Angioplasty and Drug-Eluting Stents for Infrapopliteal Lesions in Critical Limb Ischemia (PADI) Trial. Circ Cardiovasc Interv. 2016 Feb;9(2):e002376. doi: 10.1161/CIRCINTERVENTIONS.114.002376. PMID: 26861113; PMCID: PMC4753788.

Bosiers M, Scheinert D, Peeters P, Torsello G, Zeller T, Deloose K, Schmidt A, Tessarek J, Vinck E, Schwartz LB. Randomized comparison of everolimus-eluting versus bare-metal stents in patients with critical limb ischemia and infrapopliteal arterial occlusive disease. J Vasc Surg. 2012 Feb;55(2):390-8. doi: 10.1016/j.jvs.2011.07.099. Epub 2011 Dec 14. PMID: 22169682.

On the other hand, some of the most relevant studies in this field that are also systematic reviews and meta-analyses have not been included. Consider their inclusion in the discussion and interpretation of the results.

Soga Y, Takahara M, Iida O, Nakano M, Yamauchi Y, Zen K, et al. Propensity Score Analysis Comparing Clinical Outcomes of Drug-Eluting vs Bare Nitinol Stents in Femoropopliteal Lesions. J Endovasc Ther 2016;23:33–9

Meng FC, Chen PL, Lee CY, Shih CC, Chen IM. Real-World Comparison of Drug-Eluting and Bare-Metal Stents in Superficial Femoral Artery Occlusive Disease with Trans-Atlantic Intersociety Consensus B Lesions: A 2-Year, Single-Institute Study. Acta Cardiol Sin 2018;34:130–6

Zhang J, Xu X, Kong J, Xu R, Fan X, Chen J, Zheng X, Ma B, Sun M, Ye Z, Liu P. Systematic Review and Meta-Analysis of Drug-Eluting Balloon and Stent for Infrapopliteal Artery Revascularization. Vasc Endovascular Surg. 2017 Feb;51(2):72-83. doi: 10.1177/1538574416689426. Epub 2017 Jan 19. PMID: 28103754.

Ding Y, Zhou M, Wang Y, Cai L, Shi Z. Comparison of Drug-Eluting Stent with Bare-Metal Stent Implantation in Femoropopliteal Artery Disease: A Systematic Review and Meta-Analysis. Ann Vasc Surg. 2018 Jul;50:96-105. doi: 10.1016/j.avsg.2017.12.003. Epub 2018 Mar 4. PMID: 29514049.

Varetto G, Gibello L, Boero M, Frola E, Peretti T, Spalla F, Verzini F, Rispoli P. Angioplasty or bare metal stent versus drug-eluting endovascular treatment in femoropopliteal artery disease: a systematic review and meta-analysis. J Cardiovasc Surg (Torino). 2019 Oct;60(5):546-556. doi: 10.23736/S0021-9509.19.11115-9. Epub 2019 Sep 13. PMID: 31527577.

6. PLOS authors have the option to publish the peer review history of their article (what does this mean?). If published, this will include your full peer review and any attached files.

Reviewer #1: No

Reviewer #2: No

Reviewer #3: No

---

## [Author Response · Author response to Decision Letter 0]

21 Aug 2023

Dear Editor and Reviewers,

We are honored to receive your guidances and corrections regarding the manuscript we submitted. We sincerely hope that after revision, this manuscript can approach the publication requirements of the journal. The followings are our replies to the Reviewers. If there is anything inappropriate, please comment. We are very willing to further revise the manuscript. Due to the lack of profi-ciency in using English, our wording is directly translated from the Chinese context. If there is any offense, we look forward to your understanding.

Reviewer #1

Major concerns

1. As the author stated, the length of the disease is critical for the outcome, length should be ana-lyzed.

Reply: Thank you for your comment. The length of the lesion largely represents the severity of the lesion, so we believe that it affects the outcomes. But we would like to respectfully point out that the main purpose of all studies included in this manuscript was to explore whether the DES can bring better outcomes for femoropopliteal artery (FPA) diseases compared to traditional methods, and the distributions of baseline variables (including lesion length) data between groups in these studies were mostly not statistically different. The study we present aimed to quantita-tively pool their outcomes to arrive at a more convincing result. Of course, we had considered performing subgroup analyses. However, in each meta-analysis, the data of the included studies were less, which could not be further disassembled for subgroup analyses. This is really regretta-ble.

2. Debate on the popliteal artery stent implantation is still ongoing because of the anatomy, is the length of disease of the popliteal artery a risk factor?

Reply: Thank you for your comment. We agree with you that the patency of the longer FPA stents, especially those covering the middle and distal segments of popliteal artery, is worrying. As shown in the revised Table 1, 5 out of the 7 studies (with 1 new study added) included in the analyses contained the popliteal artery as one of the target lesions, while 3 of them clearly report-ed that the included popliteal arteries were all in the upper segment (above the tibial plateau). Moreover, available data showed that the proportions of isolated popliteal artery lesions that did not involve the femoral artery were all very small and not statistically significant difference be-tween groups. Like the aforementioned limitation, due to limited data, it was indeed not possible to conduct subgroup analyses based on this variable.

3. The short and long term improvement of both therapies.

Reply: Thank you for your comment. The problem that troubled us was that if the outcome meas-urement was set as the relative risk (RR) or odds ratio (OR) of cumulative incidence at different specific times (such as 1 year, 3 years, 5 years after intervention), the number of studies included in each analysis would be clearly not enough. More importantly, almost all the outcomes of FPA revascularizations are time-to-event data. Please excuse us for taking the liberty and allow us to give an example. Undoubtedly, the all-cause death (ACD) risk of a patient at 1st postoperative month is different from that at 10th postoperative year; the probability of the stent remaining primary patency (PP) varies certainly between 1st postoperative day and 2nd postoperative year. Even if we set a time frame for observation (such as only observing data within one postoperative year), the risk of outcomes for patients varies at different times during this period (such as 1st month and 12st month). Therefore, compared with RR and OR, the hazard ratio (HR) is obviously more appropriate as a measure of the outcomes of interest. This indicator incorporates the time variable into the calculation, thus eliminating the external disturbance of time. In addition, this approach can also increase the numbers of studies included in the pooled analyses, making im-possible analyses feasible.

In fact, the practice of using HR as the outcome measure for a clinical research or meta-analysis has been very common in fields such as Oncology where most outcomes are also time-to-event data. In the fields of Vascular Surgery and Interventional Radiology, such clinical studies are also gradually increasing. But for the outcomes we are interested in, we have not yet found any me-ta-analysis using HR as the outcome measure. We think this is the highlight of the presented study.

4. What are the risk factors that impact the outcome?

Reply: Thank you for your comment. This issue represents the core pursuit of clinical physicians and is indeed something we should explore. However, we respectfully point out that the presented study is a meta-analysis on whether different treatment methods have different effects on out-comes of interest, rather than that exploring other suspicious predictive factors. If regression analyses are planned to explore predictors that affect outcomes, we believe that original data based on individual patients are needed and that will be another clinical study.

5. Runoffs are critical to the endovascular therapy, the condition of the runoffs.

Reply: Thank you for your comment. We admire your point of view very much. The number and patency of the runoffs may affect the risk of restenosis. According to your request, we have added the baseline variable of runoff count to Table 1. Out of 7 included studies, 5 established technical requirements for the presence of at least 1 unobstructed runoff artery, but only 2 reported the number of runoffs without statistical differences between groups.

6. Tyro errors should be corrected. like "2000".

Reply: Thank you for your comment. We apologize for making such low-level mistake and have made the modification.

Minor concerns

1. Page 3, Line 58-63; Are there any other criteria for inclusion in the population? For example, age, gender, etc.

Reply: Thank you for your comment. We only set the inclusion criteria for the literature, but not for the patients. We hoped to expand the sample (literature) size as much as possible. If some sub-group analysis was needed, the included studies might be grouped by population characteristics or other variables, rather than initially excluding the literature that included some population.

2. Page 5, Line 143-145; Please add additional documentation to support this.

Reply: Thank you for your comment. In the included study mentioned in the manuscript, a sec-ondary randomization grouping was conducted. To avoid the presence of patients receiving DESI in both arms, we only included patients after secondary grouping. To make this point straightfor-ward, we have created an additional image (S2 Fig).

3. Page 9, Line 209; Please add literature to support that PTA alone has the disadvantage of a high restenosis rate, as well as the need for a TLR.

Reply: Thank you for your comment. We have added the corresponding reference ([4]).

4. Page 3，Line65，What specifically is meant by the failure to qualify the ending? What are the criteria for determining re-narrowing? What are the inclusion criteria for recanalization? Do dif-ferent inclusion criteria affect the results?

Reply: Thank you for your comment. When selecting literatures, we did not limit the specific definitions of the outcomes in the studies to be included. That is to say, we allowed for slight dif-ferences in the outcome definitions among different included literatures. This helped to increase the sample sizes for pooled analyses. If the inclusion criteria were too strict, many outcomes of interest would miss the opportunity to be analyzed. And we found that many studies did not pro-vide specific definitions for some outcomes. It is obviously inappropriate to exclude these studies solely based on this principle. We have presented the definitions of outcomes of interest for each included study in Table 2. Fortunately, all definitions were very close. For example, the definition of primary patency (PP) was basically based on the premise that no more than 50% of the arterial diameter was restenosis determined through imaging examinations; TLR and ISR were also roughly defined on this basis. Of course, if there were particularly outrageous definitions, we would also consider excluding them from the analysis. Admittedly, different definitions might result in some errors when pooling the effects among different studies, but we believe that this was not enough to enable us to establish stricter inclusion criteria, especially when the number of available literatures was limited.

5. Some of the results used data from only two articles, and that some of the data were extracted from the KM curves? Do the authors have a solution for this?

Reply: Thank you for your comment. Yes, we did. Extracting data from KM curves and deriving HR values is an effective method to include those literatures that did not directly report HR values into meta-analyses to improve sample sizes. This method is common in some fields where the HR is also used as a common outcome measure, such as Oncology, and has been proven to have high statistical value. We provided references in the presented manuscript. In practice, we tried to take the points denser from a curve and the calculated more frequently, so as to reduce the error. We believe that the difference between the HR value derived by this method and the actual HR value is less than 0.1, which is reliable.

6. Page 11，line 306 Did the authors consider including non-RCT results from "the new genera-tion of DES" to see if outcomes changed significantly?

Reply: Thank you for your comment. Compared with those of other controlled studies, the evi-dence level of a RCT is obviously much higher. Moreover, the quality (or bias risk) assessment criteria for other types of study are also different. Please allow us to take the liberty to say, we be-lieve that it is not feasible to mix and analyze different types of study.

Reviewer #2

1. Well done, nice workup and nice well organized results. You did a rigorous point by point re-view. In my opinion it deserves very wide exposure for a maximum benefit for specialties work-ing in this filed.

Reply: Thank you for your recognition of our study. We are very honored.

Reviewer #3

1. It would be appropriate to include the bibliographic search terms within the supplementary material.

Reply: Thank you for your recognition. We did indeed do what you said (S1 Fig).

2. There are randomized clinical trials that have not been included in the analysis, such as: (omit-ted).

Reply: Thank you for your comment. Your suggestion has been very helpful to us. The first liter-ature you pointed out is about Eluvia, a new generation DES, and we did retrieve it at the begin-ning of our study. However, due to our negligence, we initially thought that it had not reported direct HR values and convertible KM curves. However, after your suggestion, we carefully read it again and were pleased to find that it displayed a KM curve on 1-year PP in its supplementary materials. We successfully included this study in the meta-analysis for the HR values of PP. Fur-thermore, it is regrettable that the other two articles are based on patients with infrapopliteal artery disease and do not include available outcome data for the femoropopliteal artery disease of inter-est to us.

3. On the other hand, some of the most relevant studies (omitted) in this field that are also system-atic reviews and meta-analyses have not been included. Consider their inclusion in the discussion and interpretation of the results.

Reply: Thank you so much for your professional advice! These literatures have great benefits in improving the quality of our study. We have added them in appropriate locations. We would like to express our sincere gratitude to you again.

The above are all our responses. Once again, we pay tribute to the efforts of the Editor and Re-viewers.

Mingxuan Li, Haixia Tu, Yu Yan, Zhen Guo, Haitao Zhu, Jiangliang Niu, Mengchen Yin

Beijing

August 16, 2022

---

## [Decision Letter · Decision Letter 1]

29 Aug 2023

Meta-analysis of outcomes from drug-eluting stent implantation in femoropopliteal arteries

PONE-D-23-21377R1

Dear Dr. Li,

We’re pleased to inform you that your manuscript has been judged scientifically suitable for publication and will be formally accepted for publication once it meets all outstanding technical requirements.

Kind regards,

Redoy Ranjan, MBBS, MRCSEd, Ch.M., MS (CV&TS), FACS

Academic Editor

PLOS ONE

Additional Editor Comments (optional):

Review Comments to the Author

Reviewer #1: The authors have answered this reviewer's question.

And this manuscript is good for the treatment of PAD patients.

Reviewer #2: Interesting conclusions about DES, although most of the studies in-favor of DES , but yours saying the reverse.

---

## [Editor Report · Acceptance letter]

12 Sep 2023

PONE-D-23-21377R1 

Meta-analysis of outcomes from drug-eluting stent implantation in femoropopliteal arteries 

Dear Dr. Li:

I'm pleased to inform you that your manuscript has been deemed suitable for publication in PLOS ONE. Congratulations! Your manuscript is now with our production department. 

Kind regards, 

on behalf of

Dr. Redoy Ranjan 

Academic Editor

PLOS ONE